# Long-Range Allosteric Communication Modulated by Active Site Mn(II) Coordination Drives Catalysis in *Xanthobacter autotrophicus* Acetone Carboxylase

**DOI:** 10.3390/ijms26135945

**Published:** 2025-06-20

**Authors:** Jenna R. Mattice, Krista A. Shisler, Jadyn R. Malone, Nic A. Murray, Monika Tokmina-Lukaszewska, Arnab K. Nath, Tamara Flusche, Florence Mus, Jennifer L. DuBois, John W. Peters, Brian Bothner

**Affiliations:** 1Department of Chemistry and Biochemistry, Montana State University, Bozeman, MT 59717, USA; jrmattice7@gmail.com (J.R.M.); jennifer.dubois1@montana.edu (J.L.D.); 2Institute of Biological Chemistry, Washington State University, Pullman, WA 99163, USA; 3Stephenson Life Sciences Research Center, The University of Oklahoma, Norman, OK 73019, USA

**Keywords:** carbon fixation, conformational change, metalloenzyme, enzyme catalysis, acetoacetate

## Abstract

Acetone carboxylase (AC) from *Xanthobacter autotrophicus* is a 360 KDa α_2_β_2_γ_2_ heterohexamer that catalyzes the ATP-dependent formation of phosphorylated acetone and bicarbonate intermediates that react at Mn(II) metal active sites to form acetoacetate. Structural models of *X. autotrophicus* AC (XaAC) with and without nucleotides reveal that the binding and phosphorylation of the two substrates occurs ~40 Å from the Mn(II) active sites where acetoacetate is formed. Based on the crystal structures, a significant conformational change was proposed to open and close a tunnel that facilitates the passage of reaction intermediates between the sites for nucleotide binding and phosphorylation of substrates and Mn(II) sites of acetoacetate formation. We have employed electron paramagnetic resonance (EPR), kinetic assays, and hydrogen/deuterium exchange mass spectrometry (HDX-MS) of poised ligand-bound states and site-specific amino acid variants to complete an in-depth analysis of Mn(II) coordination and allosteric communication throughout the catalytic cycle. In contrast with the established paradigms for carboxylation, our analyses of XaAC suggested a carboxylate shift that couples both local and long-range structural transitions. Shifts in the coordination mode of a single carboxylic acid residue (αE89) mediate both catalysis proximal to a Mn(II) center and communication with an ATP active site in a separate subunit of a 180 kDa α_2_β_2_γ_2_ complex at a distance of 40 Å. This work demonstrates the power of combining structural models from X-ray crystallography with solution-phase spectroscopy and biophysical techniques to elucidate functional aspects of a multi-subunit enzyme.

## 1. Introduction

Atmospheric carbon dioxide is a source for building biomass in plants and certain microbes. Carboxylases catalyze CO_2_ and fixation into organic substrates. Most enzymes in this class require biotin or other cofactors [1]. Acetone carboxylases are evolutionarily distinct carboxylases that do not share sequence identity with other carboxylases and do not employ biotin as a cofactor. *X. autotrophicus* acetone carboxylase (XaAC) has been proposed to catalyze the ATP-dependent phosphorylation of acetone and bicarbonate to form intermediates that react to form acetoacetate at a mononuclear Mn(II) site.

Structures of the 360 kDa α_2_β_2_γ_2_ hexameric complex have been determined in ligand-free (PDB: 5SVC), AMP-bound (PDB: 5SVB) and AMP-acetate-bound (PDB: 5M45) forms have been determined by X-ray diffraction methods (Figure 1 and Appendix A) [2]. Surprisingly, the models revealed that the sites for nucleotide binding and presumably phosphoryl transfer and the Mn(II) sites where acetoacetate is formed are separated by ~40 Å. Significant conformational differences were observed when comparing nucleotide-bound and nucleotide-free states, mainly manifested in a rigid body reorientation of the α and β subunits relative to one another. The rigid body reorientation of the subunits results in significant differences in access to the nucleotide binding and Mn(II) sites and the opening of a 40 Å channel through the protein connecting these sites.

From these observations, it was postulated that the formation of the channel protects the reactive intermediates from hydrolytic decomposition during their migration to the Mn(II) active sites. Interestingly, it was observed that Mn(II) coordination differed in the three observed structures based on a repositioning of glutamate (αE89), which serves as a coordinating ligand in the AMP-bound condition, but not in the ligand-free or AMP + acetate structures (Figure 1C). In the latter, acetate takes the coordinating αE89 ligand’s place. These differences provided the basis for a mechanistic proposal of a mechanical couple for modulating conformational states where nucleotide binding and substrate activation within the open cleft result in the formation of the αE89-coordinated Mn(II) site, closing the cleft and opening a tunnel for the migration of the intermediates. The subsequent migration and binding of substrates at the Mn(II) site results in the displacement of the αE89 coordination, triggering conformational change back to the substrate-binding mode without the tunnel but with the open cleft for the binding of nucleotides and substrates for another round of catalysis.

In this work, we have experimentally probed the dynamics and conformation changes associated with catalysis by interrogating the native enzyme and site-specific amino acid substituted variants in poised catalytic states using a combination of electron paramagnetic resonance (EPR) and hydrogen/deuterium exchange coupled with mass spectrometry (HDX-MS). Our results demonstrate reciprocal communication between the sites of formation of phosphorylated intermediates and the sites of acetoacetate formation is mediated by exchangeable carboxylate ligand αE89 at the Mn(II) sites.

## 2. Results

### 2.1. Substitution of αE89 Confirms an Essential Role in Catalysis

To probe the role of αE89 in the catalytic cycle, ATP hydrolysis and the production of ADP, AMP, and acetoacetate were monitored in AC variants in which α-subunit glutamate 89 was substituted by aspartate (αE89D) and alanine (αE89A). The hypothesis for these substitutions was that αE89D should still be able to coordinate Mn(II) in the same manner as αE89 and would retain some activity. However, αE89A is incapable of coordinating Mn(II), and as such, we anticipate the complete loss of activity if coordination is essential. To test this, acetoacetate production was measured using a coupled assay with β-hydroxybutyrate dehydrogenase (β-HBDH), which converts acetoacetate to 3-hydroxybutanoate in the reverse reaction upon the oxidation of NADH [3]. Parallel assays were conducted to analyze acetoacetate production alongside ATP hydrolysis and ADP and AMP production. For AC, the rate of ATP consumption is 76.3 µM/min, and the acetoacetate formation is 33.5 µM/min. The αE89D variant showed comparable activity, with a rate of ATP consumption of 89.2 µM/min and acetoacetate formation of 41.6 µM/min. The αE89A variant produced no acetoacetate and had no ATPase activity above the non-productive baseline measured for all three enzyme forms without acetone. The loss of acetone-dependent ATP consumption and final product formation in αE89A suggests that carboxylate coordination of the Mn(II) by residue 89 is required for both activities.

### 2.2. EPR Studies Confirm the Role of αE89 in Mn(II) Coordination

X-band EPR was used to monitor the changes in Mn(II) coordination that accompany AMP binding to XaAC in the wild-type enzyme, the conservative αE89D variant, and the αE89A variant lacking the carboxylate to coordinate Mn(II), specifically in the AMP-bound XaAC structure. As demonstrated in previous studies, XaAC exhibits a low-intensity EPR signal at g = 2 that converts to a well-defined six-line hyperfine signal upon adding AMP [4]. This signal is reminiscent of spectra for well-characterized high-spin (S = 5/2) Mn(II) complexes with nitrogen and oxygen ligands in octahedral symmetry (Figure 2) [4,5]. The changes in the EPR spectra before/after AMP binding to the enzyme are consistent with differences in the Mn(II) coordination environment observed in X-ray crystal structures. The Mn(II) in the ligand-free XaAC is in a five-coordinate state with H150, H175, a bidentate carboxylate D153, and an aquo/hydroxo species as ligands. In the AMP-bound XaAC, the Mn(II) has a more symmetric six-coordinate environment, including the nitrogen ligands from the H150 and H175 amines and two bidentate carboxylate ligands of D153 and E89 [4,5]. We hypothesized that the αE89D XaAC would be capable of an analogous shift from the initial five- to the six-coordinate geometry in the AMP-bound state more or less identical to that of the wild-type, while Mn(II) in the αE89A variant would remain in the more disordered five-coordinate state. The αE89D and αE89A XaACs exhibited an EPR spectrum with features near g = 2 similar to those measured for substrate-free wild-type XaAC at comparable protein concentrations. A parallel analysis of αE89D and αE89A indicated that αE89D XaAC undergoes analogous changes to the WT XaAC in the EPR spectra, while αE89A XaAC does not (Figure 2). It should be noted that an earlier investigation of XaAC confirmed that the Mn(II) coordination is not due to a direct interaction with ATP or AMP in proteins [4] or free Mn(II) [6] This provides the first direct evidence linking changes in Mn(II) EPR spectra to E89 coordination.

### 2.3. Thermal Stability Shifts with Substrates Bound to both WT and αE89A

To measure the overall stability of AC and better understand the impact of ligand binding and mutation, we employed Differential Scanning Fluorimetry (DSF) [7]. Analysis of WT and the αE89A variant poised at different points in the catalytic cycle was carried out (Figure 3). In WT, substrate binding shifts the T_m_ from 73.5 °C in the unbound protein to 76.5 °C with AMP-bound and 76 °C for AMP-acetate-bound. Both substrate-bound forms have a temperature transition that occurs before global denaturation. The αE89A variant is significantly destabilized, having a T_m_ that is almost 20 °C less. However, the binding of AMP or AMP-acetate restores the Tm essentially to WT values and the lower temperature transition is present. The ligand-induced changes in fluorescence are consistent with an allosterically controlled complex and further point to αE89 as an important switch.

### 2.4. Ligand Binding Alters the Hydrogen Bonding Network of AC

HDX-MS can provide insights into the effects of substrate binding and is a powerful tool for investigating protein conformational change and dynamics, particularly in large complexes [8,9]. Based on the significant differences in the structural models for the unbound and ligand-bound forms of XaAC, we anticipated differences in the exchange pattern due to changes to the hydrogen bonding network and dynamics in the three subunits (α, β, γ) [2]. To initiate exchange, AC was diluted 10-fold with D_2_O-containing buffer and allowed to exchange for 1, 8, 20, 60, 180, and 1440 min before Liquid Chromatography Mass Spectrometry (LCMS). Deuterium uptake was measured at each time point for unbound, AMP-bound, and AMP-acetate-bound conditions. The AMP-bound state captures the conformation where E89 coordinates the Mn(II). AMP-acetate captures the product release structure where acetate displaces E89, which is rotated away from the active site Mn(II). Analysis of the three experimental conditions showed large differences in deuterium uptake for the β-subunit, while the α and γ subunits did not show distinct changes upon substrate binding (Appendix A). In the β-subunit, adding AMP resulted in greater deuterium uptake across all time points compared to the ligand-free XaAC. The combination of AMP and acetate caused even more deuterium uptake.

Next, we repeated the intact protein HDX-MS on αE89A, using the same conditions as for WT XaAC. Unlike WT AC, in which significant differences in deuterium uptake for the intact proteins were only observed for the β-subunit, αE89A had distinct changes in deuterium incorporation in all three subunits upon ligand binding (Appendix A). For the α and γ-subunits, adding AMP-acetate decreased uptake, while greater exchange was observed for the β-subunit. These results show that the αE89A variant has an altered hydrogen bonding network with local and distal effects, confirming αE89 as a key link in the mechanical properties of the complex. The difference in deuterium uptake between AMP and AMP-acetate supports the structural data that the binding of acetate and AMP favors a unique conformation [2,10]. In addition, the HDX-MS, EPR, and DSF data are consistent with a model in which the catalytic cycle of XaAC is allosterically controlled by ligand-induced changes.

### 2.5. Mechanical Connectivity Between the Substrate Binding and Mn(II) Sites

Pepsin digestion was added to the experiment to identify specific regions of XaAC that have altered hydrogen bonding stability at different points in the catalytic cycle. This additional step facilitates measuring changes in solvent accessibility and/or dynamics of specific peptides correlated with catalysis. Using the same conditions (unbound, AMP, AMP-acetate), peptide-level HDX was measured at 0.5, 3, 30, 180, and 1440 min to capture fast- and slow-exchanging peptides. Peptide mapping of XaAC after digestion with pepsin provided a sequence coverage of 95% for the α subunit, 96% for β, and 85% for γ (see Appendix A for the complete HDX data). We focused our initial analysis on eight peptides (α85–96, α125–134, α412–422, α442–455, α523–536, β117–130, β525–532 and β677–684) representative of key locations including substrate binding, proximity to the α/β subunit interface, the Mn(II) active site, and the 2-fold interface between αβγ trimers. We first looked at how AMP binding impacted exchange. At the 3 h time point α125–134, α523–536, β525–532 and β677–684 exhibited less exchange upon AMP binding, while α85–96, α412–422, α442–455, β117–130 had more exchange (Figure 4 and Appendix A). However, when AMP and acetate were both present, seven of the eight peptides displayed opposite exchange patterns. Given the spatial distribution of these peptides, these data demonstrate that large-scale changes in the stability and dynamics of the hydrogen bonding network occur in the XaAC complex that are germane to the catalytic cycle.

Peptide α85–96, which contains the Mn(II) coordinating sidechain of αE89 had greater deuterium uptake in the AMP-bound condition compared to the unbound and less with the AMP-acetate-bound condition. This pattern of exchange is consistent with a helix-to-coil transition as observed in the AMP-bound structure of XaAC that positions αE89 to coordinate Mn(II) [2]. The αE89 carboxylate does not coordinate the Mn(II) in the AMP-acetate-bound enzyme and the α85–96 peptide resumes its α-helical structure. Analysis of α85–96 served as a positive control that the protein and experiment were working properly as E89 does not coordinate Mn(II) in unbound or AMP-acetate conditions, and the structural models show that α85–96 forms a helix which is expected to decrease exchange. However, because much less exchange occurred in the AMP-acetate condition compared to the unbound condition, there must be differences in conformation and/or dynamics that are not revealed in the available X-ray crystal structures. Two of the highlighted peptides are proximal to the 2-fold interface between the αβγ trimers of the XaAC hexamer. The exchange for α442–455 increases with AMP and decreases with AMP-acetate, while the other interface peptide, α125–134, decreases in both conditions compared to unbound. The fact that both of these peptides show decreased exchange when AMP and acetate are bound suggests a stabilization of the interface at this point in the catalytic cycle.

### 2.6. αE89A Variant Has Local and Distal Changes in HDX Pattern

The EPR analysis and catalytic assays indicated that Mn(II) coordination by the αE89 side chain is critical for acetoacetate production. This suggests that conformational shifts in αE89 are part of a mechanical couple between the Mn(II) and nucleotide binding sites of XaAC. If αE89 is an important link, disrupting the mechanical chain should alter the stability, dynamics, and communication in distal regions of XaAC. To test this, we performed peptide-level HDX on the αE89A variant, testing unbound, AMP, and AMP-acetate. It was immediately evident that eliminating the carboxylate sidechain altered the allosteric behavior.

In the β-subunit of the αE89A complex, the deuterium uptake of AMP and AMP-acetate-bound forms was less than the ligand-free form for peptides β117–130 and β677–684 (Figure 4). All six of the other peptides, including the peptide with αE89A, exhibited more exchange with AMP and AMP-acetate. We also noted that all of the highlighted peptides from the α subunit showed greater rates of exchange when AMP was added. This was unexpected because the nucleotide binding site is in the β subunit. These results demonstrate that the single amino acid substitution disrupted the long-distance mechanical coupling and caused widespread changes in the stability of the global hydrogen bonding network.

## 3. Discussion

The extraordinary conformational flexibility of carboxylate ligands [11], which can shift between monodentate, bidentate, metal-bridging, and nonbonding modes during a single catalytic cycle, makes them essential components of diverse metalloenzyme active sites. This flexibility drives reactions at the binuclear iron sites of ribonucleotide reductase and soluble methane monooxygenase, where computational methods suggest that carboxylate switching between bridging and terminal monodentate binding modes during the catalytic cycle is both thermodynamically favorable and rapid [12,13,14]. Carboxylate shifts have also been successfully incorporated into the design of biomimetic model complexes that simulate metalloenzyme active sites [15,16]. The dynamic nature of the biomimetics was demonstrated spectroscopically during reactions at the metal center [17]. In each case, carboxylate shifts steer catalysis in the immediate coordination sphere of the metal.

In contrast to these well-established paradigms, structural analyses of XaAC suggested a carboxylate shift that couples both local and long-range structural transitions. We hypothesized that shifts in the coordination mode of a single carboxylic acid residue (αE89) mediate both catalysis proximal to a Mn(II) center and communication with an ATP active site in a separate subunit of a 360 kDa α_2_β_2_γ_2_ complex at a distance of 40 Å [2]. The observed elimination of acetoacetate production and the changes in the EPR signal for the αE89A variant and restoration in αE89D provides direct evidence that carboxylate coordination of Mn(II) by αE89 is an essential part of the catalytic cycle. In addition, this demonstrates long-distance communication between the ligand binding sites in the β-subunits and the Mn(II) active sites for acetoacetate formation.

Intact protein HDX revealed more deuterium uptake in the β-subunit upon AMP-binding, which was further enhanced with the addition of acetate (Appendix A). Acetate displaces the E89 carboxylate at the Mn(II) active site, which is in the α-subunit. The observed responses indicate bidirectional communication between the nucleotide binding sites and the Mn(II) sites of acetoacetate formation. The higher resolution of peptide-level HDX provided information on the stability of the hydrogen bonding network in specific regions of each subunit and supported mechanical coupling between the sites. Peptides in α- and β-subunits had opposing changes in exchange upon AMP vs. AMP-acetate binding (Figure 4). Of special interest is Peptide α85–96, which contains αE89. Exchange increased as the α-helix unfolded upon AMP binding. In the presence of AMP and acetate, there was less exchange in this peptide consistent with helix formation. The same action was present in peptides α412–422 and α442–455. The three other marker peptides, (α523–536, β525–532, β677–684) all had the opposite response: decreased exchange with AMP and increased exchange with AMP-acetate. The peptide level HDX shows there are clear changes in subunit conformation and dynamics mediated by ligand binding at sites up to 40 Å away and across protein interfaces. Taken together, these data shows that upon acetate binding, the region near the Mn(II) and the interface between αβγ halves of the heterohexamer is stabilized, while at the same time the nucleotide binding site and the α and β subunit interfaces become more dynamic (Figure 5). Importantly, the altered dynamics occur in conjunction with conformational changes associated with different steps in the catalytic cycle and opening of a substrate channel. The HDX data also confirms that αE89 plays a significant role in tuning dynamics near the αβγ trimer interface and is a critical component of a mechanically coupled pathway that connects the active sites.

We determined that a carboxylate at the αE89 position is essential for the acetoacetate-forming reaction catalyzed at the Mn(II) site. The αE89D variant (ethanoate) generates product at a similar rate. The αE89A variant (methyl) does not produce acetoacetate, but is capable of binding Mn(II) and hydrolyzing ATP at the same basal rate as wild-type XaAC. Data presented here suggests that the αE89 carboxylate cycles between free and Mn(II) coordinated during catalysis. Our proposed mechanism has each of the two organic co-substrates in their phosphoryl-activated, carboxyphosphate and phosphoenolacetone forms bind to the Mn(II) active site in a manner that displaces αE89 coordination (Figure 6). The free αE89 carboxylate could act as an active site base toward the carboxyphosphate, which has been shown computationally to have an exceptionally stable intramolecular charge-assisted hydrogen bond between the oxygen atoms of the phosphoryl and carboxy moieties [14]. Or, it could form an essential link involving the intermediates in a hydrogen bond network. The αE89 side chain rotates into bidentate coordination when the enzyme is in the AMP product-bound state, leaving a catalytically unavailable 6-coordinate Mn(II) at the end of the reaction cycle, after which AMP departs and the initial 5-coordinate Mn(II) reforms. The current work indicates that the role of αE89 extends beyond these proposed roles in catalysis and acts as a trigger to promote the aforementioned conformational changes that link the sites for substrate activation to the Mn(II) sites of C-C bond formation via a conduit that encapsulates reactive intermediates.

We envision a toggling between a substrate activation state characterized by a cleft that binds ATP, acetone, and bicarbonate that then react to form intermediates and that these phosphoryl transfer steps transduce conformational changes 40 Å to the Mn(II) that result in αE89 carboxylate coordination and the formation of a tunnel that protects and directs the intermediates to the Mn(II) site. The binding of intermediates then displaces the αE89 carboxylate coordination, freeing the side chain participate in catalysis and at the same time effecting the transduction of reciprocal conformational change that resets the system for another round of catalysis upon product release.

The opening of an interior channel to encapsulate a pair of unstable intermediates is evocative of an emerging pattern by which nature controls CO and CO_2_ gases in prokaryotes. Acetogenic bacteria, for example, assimilate CO_2_ as a carbon and energy source using CO dehydrogenase (CODH) in complex with acetyl CoA synthase (ACS) [18,19,20]. The nickel-iron center in CODH catalyzes the reduction of CO_2_ to CO, which subsequently migrates through a 70 Å hydrophobic channel to the acetyl CoA synthetase, where it condenses with the thiol group of CoA. Methanogenic archaea use CODH and ACS as modules in the multienzyme acetyl-CoA decarbonylase/synthase (ACDS) complex, in which CO_2_ is reduced completely to methane [21]. Cyanobacteria have evolved carboxysomes, protein-based microcompartments that encapsulate their primary enzyme for carbon fixation from CO_2_, D-ribulose 1,5-bisphosphate carboxylase/oxygenase (RubisCO). These semi-permeable compartments screen out O_2_ and permit regulated entry of CO_2_ from a cytoplasmic pool of bicarbonate and the enzyme carbonic anhydrase [22]. Each of these macromolecular systems has a mechanism for entrapping gaseous and/or unstable intermediates, channeling them toward desired co-substrates, while coordinating the production of intermediates at multiple active sites. AC is remarkable for housing each capability within one protein multimer with a uniquely efficient carboxylate switching mechanism.

The biophysical analysis of XaAC at different steps in the catalytic cycle has revealed the coordinated changes that facilitate metabolic channeling between substrate binding sites and product formation. Similar conformational calisthenics have been proposed for tryptophan synthase (TS) [23] and guanosine monophosphate synthetase (GMPS) [24], both of which are 2-fold symmetric hetero-multimers that make use of molecular tunnels to protect reactive intermediates in a multistep catalytic process. By combining EPR and HDX, we have tracked changes in the hydrogen bonding network intimately associated with conformational changes 40 Å apart at the two active sites in each half of the complex. Importantly, the data show that nucleotide binding stabilizes distal regions of the XaAC complex while increasing dynamics near the αβγ 2-fold interface. As the catalytic cycle progresses and substrates move to the Mn(II) site, where the product acetoacetate is formed, the 2-fold interface is stabilized, and the distal regions become more dynamic. Structural studies of TS and GMPS also show the movement of loops and domains as these enzymes progress through their catalytic cycles. We have now taken this a step further with XaAC by elucidating a molecular switch that controls a helix–coil transition as well as globally tuning the stability of the hydrogen bonding network across the complex. While XaAC is an evolutionarily distinct carboxylase, we propose that the mechanical coupling and dynamical tuning are shared properties in symmetric hetero-multimeric enzymes and are fundamental to multi-step catalysis.

## 4. Materials and Methods

### 4.1. Expression and Purification of Recombinant Acetone Carboxylase

WT, E89A and E89D XaAC were expressed and purified as previously described [2]. The *E. coli* cells were lysed by sonication in 25 mM Tris, pH 7.6, 0.1 mM EDTA, and 20% glycerol. The lysate was applied to a His-NTA column (Bio-Rad, Hercules, CA, USA) and eluted with a linear gradient with an imidazole buffer (25 mM Tris, pH 7.6, 0.1 mM EDTA, 20% glycerol, and 400 mM imidazole). Fractions containing AC were pooled and buffer exchanged into a 20 mM Tris, pH 7.6 buffer and applied to a Q-sepharose (Bio-Rad) column. The protein was eluted with a linear gradient with a NaCl buffer (20 mM Tris, pH 7.6, 800 mM NaCl). Fractions were pooled and buffer exchanged into a 50 mM Tris, pH 7.6, 150 mM NaCl, 10% glycerol storage buffer and stored at −80 °C. Purified complexes were tested by size exclusion chromatography and native gel analysis to ensure integrity (Appendix A). Protein concentrations were determined by Bradford assays.

### 4.2. EPR Spectroscopy of WT and Variant AC

EPR samples of as-purified WT, E89A, and E89D *X. autotrophicus* ACs were diluted to 36 mg/mL in 50 mM Tris, pH 7.6, 150 mM NaCl buffer. For AMP containing samples, 10 mM AMP and 1 mM MgCl_2_ were added. For metal-depleted samples, 900 µL aliquots of EDTA/TEA treated AC were removed (after buffer exchanging into the AMP-free buffer to remove excess EDTA/TEA) and concentrated to 250 µL. Final concentrations were 27 mg/mL (*X. autotrophicus*) 73 mM protein. For MnCl_2_ reconstituted samples, 900 µL aliquots of MnCl_2_ treated AC were removed and buffer exchanged against metal-free 50 mM Hepes, pH 7.6, 150 NaCl with or without 1 mM AMP and concentrated to 250 µL. Final concentrations for *X. autotrophicus* AC were 20 mg/mL (AMP present) and 21 mg/mL (AMP-free), or 55 and 57 mM protein, respectively.

Samples were loaded into quartz EPR tubes and immediately flash frozen in liquid nitrogen. X-band EPR spectra were measured using an EMX continuous-wave (CW) spectrometer (Bruker Inc., Billerica, MA, USA) equipped with a continuous helium flow Oxford cryostat set to 12 K. Normal EPR parameters were: modulation amplitude 10 G, modulation frequency 100 kHz, 2 mW power, and microwave frequency 9.38 GHz. Spectra for the Mn(II)-bound AC samples were re-measured under identical conditions except for 10 mW power, which we had previously determined was well below the power at which spectral intensity saturated. Spin quantification was carried out by double-integration of this spectrum and comparison of the resulting peak area to a standard curve, generated using varying known concentrations of [Mn(II)(acetoacetate)2(1-methylimidazole)2].

### 4.3. Enzymatic Assays of WT and Variant AC

As-purified WT, E89A, and E89D AC was assayed for acetoacetate production via a couple assay with β-hydroxybutyrate dehydrogenase (β-HBDH) which converts acetoacetate to 3-hydoxybutanoate in the reverse reaction upon the oxidation of NADH, as previously described [3]. The rate of acetoacetate is proportional to the rate of NADH oxidization which is observed at 340 nm. Parallel assays were conducted to analyze acetoacetate production alongside ATP hydrolysis and ADP and AMP production. In a rubber-stoppered UV-vis quartz cuvette, the reaction components were added to a final concentration of 100 mM Tris, pH 7.6, 80 mM KCl, 54 mM KHCO_3_, 1 mM MgCl_2_, 343 µM NADH, 2 mM acetone, 0.25 mg/mL AC and 0.05 mg/mL β-HBDH and incubated at 37 °C for 4 min prior to initiation by the addition of 2 mM ATP. Once ATP was added, the cuvette was inverted to mix and a 200 µL aliquot was removed and quenched with 10 mM EDTA (final concentration), pH 8.0 while the cuvette was inserted into a Cary 6000i for UV-vis analysis at 340 nm. The reaction was allowed to proceed for 1.5 min, upon which another 200 µL aliquot was removed and quenched with 10 mM EDTA. The 200 µL aliquots were placed in a 100 kDa MWCO spin concentrator (pre-chilled on ice) and spun at 14,000× *g* to separate protein from reaction products. The flow-through was then injected onto an HPLC (Agilent Technologies, Santa Clara, CA, USA) and run with a 3% methanol (buffer B) and 97% 20 mM ammonium acetate, pH 4.5 buffer (buffer A) method for 15 min to separate the peaks. ATP, ADP, and AMP eluted at 2.2, 2.9, and 5.8 min, respectively. ATP, ADP, and AMP were quantified using a standard curve and integration of the peaks. Trials were performed in triplicate.

### 4.4. Intact Protein HDX of WT and Variant AC

Stock solutions of WT AC (44.8 mg/mL) and E89A AC (41.7 mg/mL) were mixed with MgCl_2_ (1 mM) in the presence and absence of AMP (10 mM) or AMP-acetate (10 mM). Reactions were then diluted 1:10 into deuterated reaction buffer (100 mM Tris, 100 mM KCl, pH 7.6) Control samples were diluted into a non-deuterated reaction buffer. At each time point (1, 8, 20, 60, 180, 1440 min) 10 µL of the reaction was removed and direct injected into LCMS. LCMS analysis of intact protein was completed on a 1200 infinity HPLC (Agilent Technologies, Santa Clara, CA, USA) coupled to a MicroTOF mass spectrometer (Bruker Inc., Billerica, MA, USA). Proteins were injected on a size exclusion column (Phenomenex Biozen SEC 150 × 4.8 mm) (Phenomenex Inc., Torrence, CA, USA) and infused into mass spectrometer using a flow rate of 500 μL/min with solvent conditions 90% solvent A [0.1% FA (Sigma, Burlington, MA, USA) in water and 10% solvent B (0.1% FA in acetonitrile) (Thermo Fisher Scientific, Waltham, MA, USA). Data were acquired at 2 Hz over the scan range 50 to 1700 *m*/*z* in positive mode. Electrospray settings were as follows: nebulizer set to 3.7 bar, drying gas at 8.0 L/min, drying temperature at 350 °C, and capillary voltage at 3.5 kV. Data processing was carried out with Bruker Data Analysis software (https://www.bruker.com/en/products-and-solutions/mr/nmr-software/analysis-software.html, accessed on 16 June 2025).

### 4.5. Peptide-Level HDX of WT and Variant AC

Stock solutions of WT AC (44.8 mg/mL) and E89A AC (41.7 mg/mL) were mixed with MgCl_2_ (1 mM) in the presence and absence of AMP (10 mM) or AMP-acetate (10 mM). Reactions were then diluted 1:10 into deuterated reaction buffer (100mM Tris, 100mM KCl, pH 7.6) Control samples were diluted into a non-deuterated reaction buffer. At each time point (0.5, 3, 30, 180, 1440 min) 10 µL of the reaction was removed and quenched by adding it to 60 µL of 0.75% formic acid (FA, Sigma, Burlington, MA, USA) and 0.25 mg/mL porcine pepsin (Sigma, Burlington, MA, USA) at pH 2.5 on ice. Each sample was digested for 2 min with vortexing every 30 s and then flash-frozen in liquid nitrogen. Samples were stored in liquid nitrogen until the LCMS analysis. The LCMS analysis of AC was completed as described (Patterson et al. 2020) [25]. Briefly, the LCMS analysis of AC was completed on a 1290 UPLC series chromatography stack (Agilent Technologies, Santa Clara, CA, USA) coupled with a 6538 UHD Accurate-Mass QTOF mass spectrometer (Agilent Technologies, Santa Clara, CA, USA). Peptides were separated on a reverse phase column (Onyx Monolithic C18 column, 100 × 2 mm, Phenomenex Inc., Torrence, CA, USA) at 1 °C using a flow rate of 500 μL/min under the following conditions: 1.0 min, 5% B; 1.0 to 9.0 min, 5 to 45% B; 9.0 to 11.8 min, 45 to 95% B; 11.8 to 12.0 min, 5% B; solvent A = 0.1% FA (Sigma, Burlington, MA, USA) in water (Thermo Fisher Scientific, Waltham, MA, USA) and solvent B = 0.1% FA in acetonitrile (Thermo Fisher Scientific, Waltham, MA, USA). Data were acquired at 2 Hz over the scan range 50 to 1700 *m*/*z* in the positive mode. Electrospray settings were as follows: the nebulizer set to 3.7 bar, drying gas at 8.0 L/min, drying temperature at 350 °C, and capillary voltage at 3.5 kV. Peptides were identified as previously described (Berry et al. 2018) [26] using MassHunter Qualitative Analysis, version 6.0 (Agilent Technologies, Santa Clara, CA, USA), Peptide Analysis Worksheet version 2000.06.08 (ProteoMetrics LLC, Cupertino, CA, USA), and PeptideShaker, version 1.16.42, paired with SearchGUI, version 3.3.16 (Vaudel et al. 2015), https://compomics.github.io/projects/peptide-shaker (accessed on 16 June 2025) [27]. Deuterium uptake was determined and manually confirmed using HDExaminer, version 2.5.1 (Trajan Scientific and Medical, Ringwood, VIC, Australia). Heat maps were created using MSTools (Kavan et al. 2011), https://peterslab.org/MSTools/DrawMap/DrawMap.php (accessed on 16 June 2025) [28].

### 4.6. Differential Scanning Fluorimetry

Reactions contained 2 μg of protein in 100 mM Tris, 100mM KCl, pH 7.6. WT and αE89A variant acetone carboxylase either alone or poised with ligands (10 mM AMP or 10 mM AMP-acetate) were mixed with 1.2 mL of 50× stock SYPRO orange dye (Invitrogen, Carlsbad, CA, USA) for a total reaction volume of 30 μL. The samples were loaded into a Roto-Gene Q instrument (Qiagen, Hilden, Germany) with SYPRO orange absorbance monitored at 570 nm as the temperature was ramped from 25 to 95 °C at 1 °C/min.

## Figures and Tables

**Figure 1 ijms-26-05945-f001:**
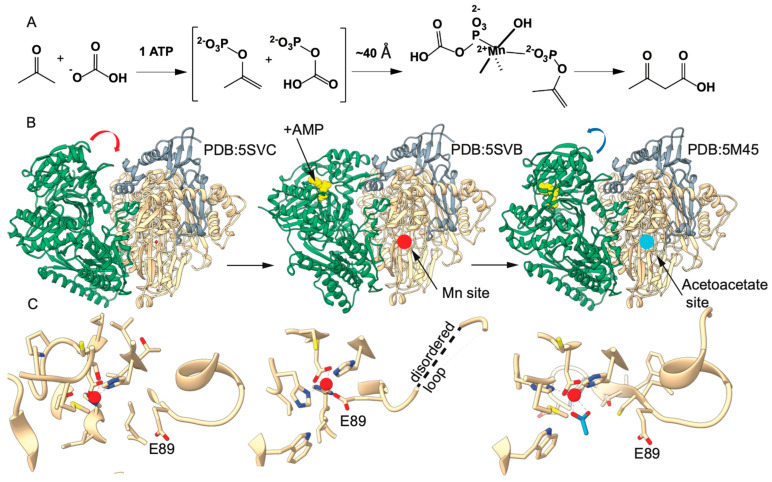
Overview of the acetone carboxylase reaction and structure. (**A**) Truncated mechanism of acetone carboxylase showing acetone and bicarbonate becoming acetoacetate. (**B**) One half of the α_2_β_2_γ_2_ hexamer of acetone carboxylase, beta subunit (green), alpha subunit (tan) and gamma (grey). Crystal structures of unbound (PDB: 5SVC), AMP-bound (PDB: 5SVB), and AMP-acetate-bound (PDB: 5M45) have been solved and depict conformational change upon substrate binding (AMP depicted in yellow) and acetate (depicted in blue). (**C**) Detailed view of E89 changing coordination during reaction. The alpha helix associated with αE89 becomes a disordered loop when coordinated to Mn(II) and reforms upon acetate coordination.

**Figure 2 ijms-26-05945-f002:**
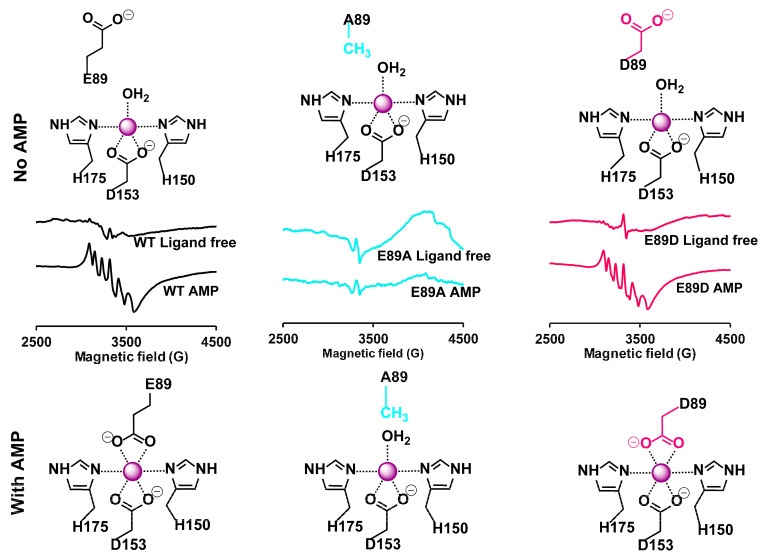
EPR analysis of the active site Mn(II). (**Left**) AC contains a Mn(II) signal at g = 2. The addition of AMP converts the weak, poorly featured EPR spectrum to a defined, six-line spectrum consistent with hyperfine coupling of the Mn(II) d-electrons with the manganese nuclear spin (I = 5/2). Mn(II) coordination without and with AMP addition are shown above and below the EPR data, respectively. (**Center**) In the αE89A variant, analogous spectral changes are not observed with the addition of AMP, as predicted for an active site without a sixth coordinating ligand. (**Right**) αE89D and the WT AC respond similarly to AMP binding. The EPR signal converts to a six-line spectrum with well-defined hyperfine coupling when AMP is bound.

**Figure 3 ijms-26-05945-f003:**
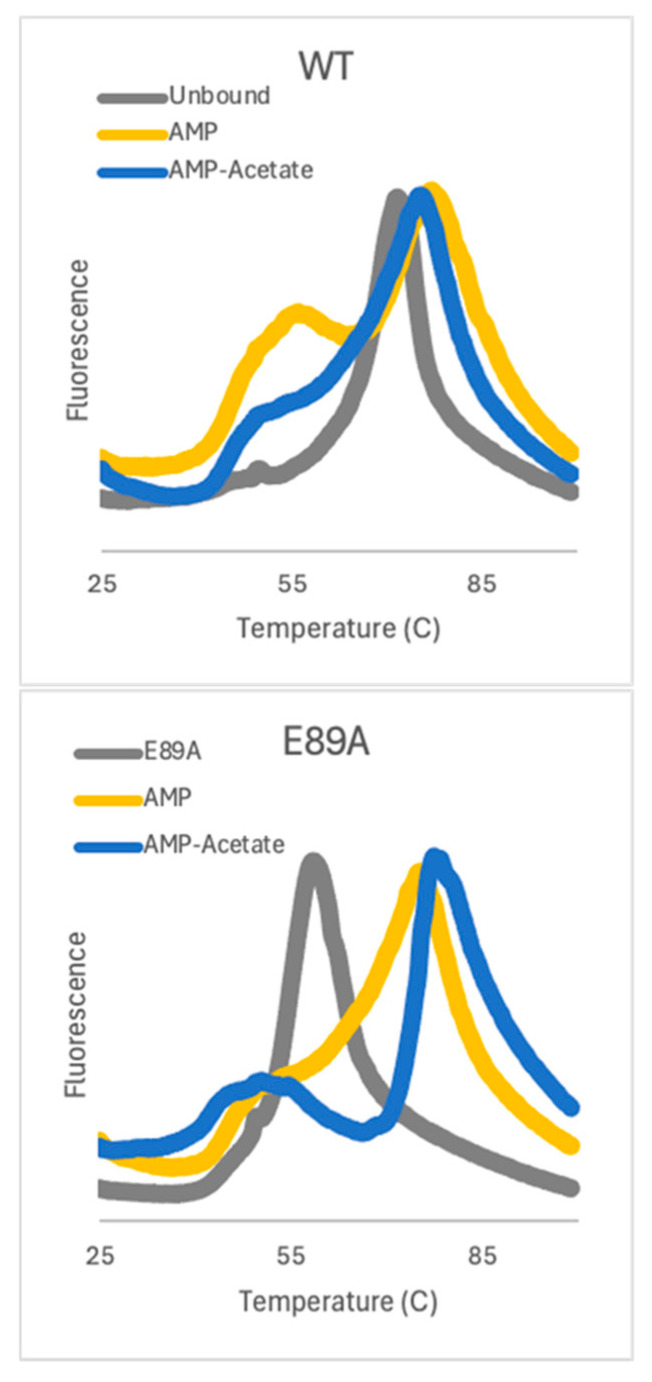
Differential Scanning Fluorimetry of WT and E89A variant. WT protein exhibited a T_m_ of 73.5 ± 0.3 °C in the unbound condition, 76.5 ± 0.5 °C with AMP-bound and, 76 ± 0.2 °C with AMP-acetate-bound. The αE89A variant exhibited a T_m_ of 59 ± 0.2 °C in the unbound condition, 76 ± 0.4 °C in the AMP-bound, and 78 ± 0.2 °C with AMP-acetate-bound condition. (n = 3 for all measurements).

**Figure 4 ijms-26-05945-f004:**
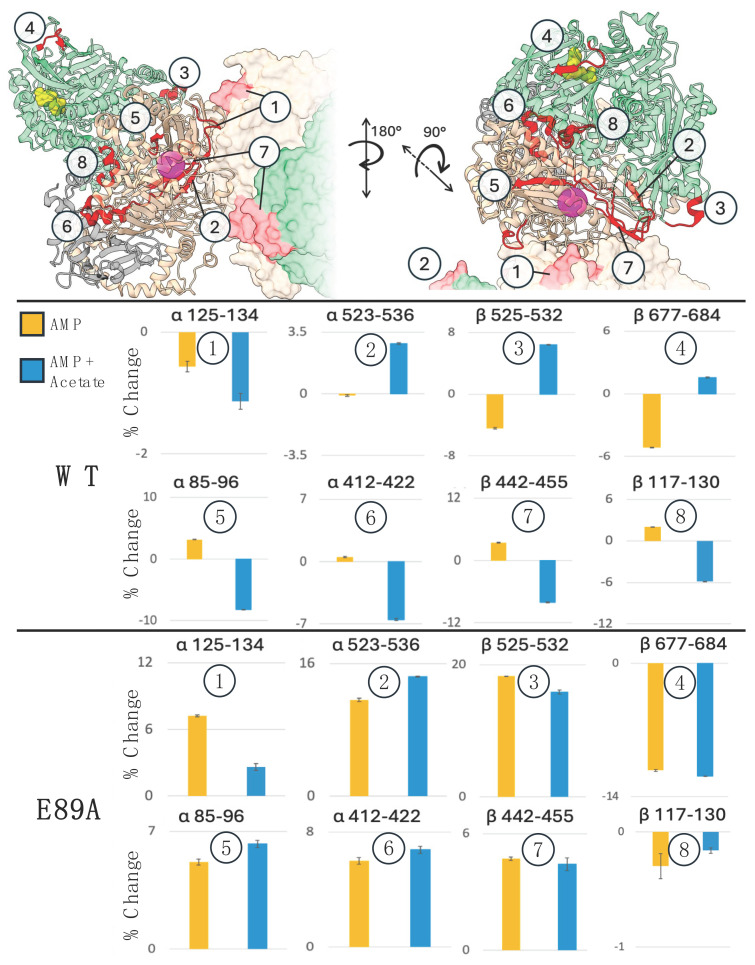
Peptide-level HD exchange shows specific changes upon ligand binding. One half of the XaAC hexamer is shown with the alpha subunit (tan), beta subunit (green), and gamma (gray). The nucleotide binding site is shown with AMP (yellow) and the Mn(II) active site is in magenta. Histograms of % change in deuterium uptake of AMP and AMP-acetate compared to unbound are shown for key peptides in red. Wild-type and αE89A have different responses to ligand binding both proximal and distal to the binding sites. Error bars show the standard deviation (n = 3).

**Figure 5 ijms-26-05945-f005:**
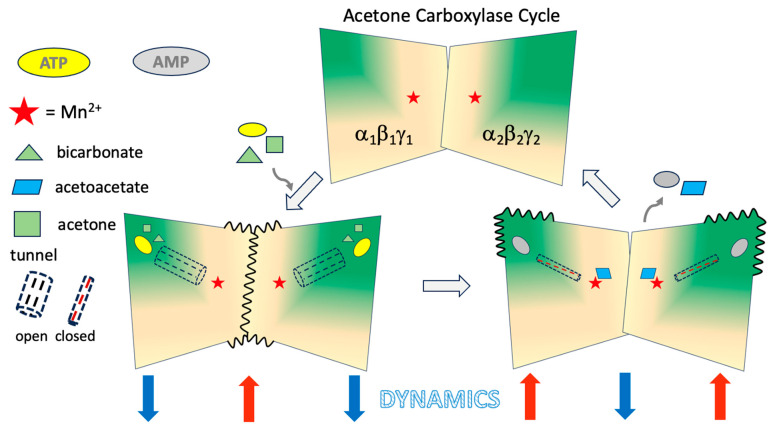
Acetone Carboxylase Cycle. Dynamic coupling in XaAC facilitates metabolic channeling and bidirectional communication between the nucleotide binding site and the Mn^2+^ active site. Binding of substrates in the beta subunit triggers a conformational change opening a molecular tunnel that allows the phosphorylated intermediates to access the Mn^2+^ active sites. αE89 acts as a molecular switch controlling allosteric communication and dynamics. Bottom left: Substrate binding leads to a decrease in dynamics of distal regions (blue arrows) of the α_2_β_2_γ_2_ hexamer and an increase around the 2-fold axis of symmetry (red arrow). Bottom right: Displacement of αE89 at the Mn^2+^ site by the phosphorylated intermediates flips a carboxylate switch, stabilizing the 2-fold axis while increasing dynamics distally to facilitate product release and another round of substrate binding.

**Figure 6 ijms-26-05945-f006:**
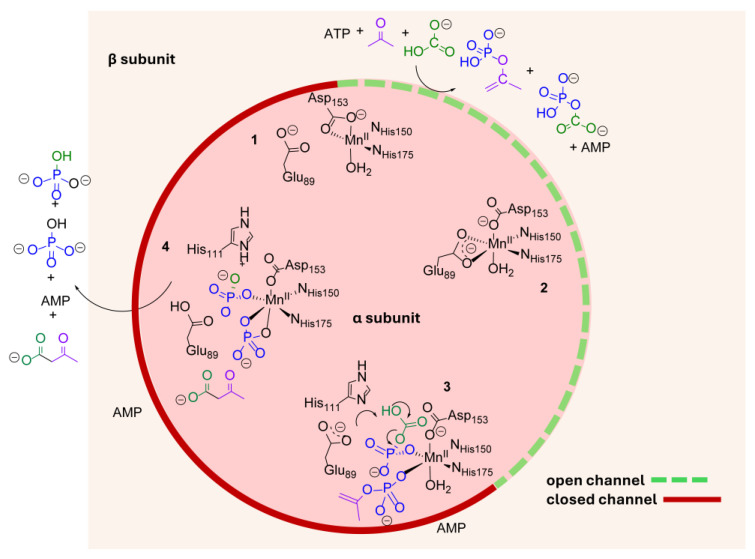
Proposed mechanism for bicarbonate-acetone coupling via acetone carboxylase. Steps occurring at the Mn(II) site (α subunit) are depicted inside the circle. Steps occurring at the ATP-binding active site (β subunit) are in the area shaded tan. A red solid line indicates that the channel connecting the two sites is proposed to be closed, while the dashed green line indicates that it is open. In this model, ATP binding triggers E89 coordination to Mn(II). E89 coordination to Mn(II) controls channel opening, in turn. Species along the pathway include: (1) ligand-free AC (resting state) with a 5CN Mn(II) and closed interior channel; (2) three substrates bind, phosphoryl transfer follows, and the phosphorylated intermediates to migrate to a 6CN Mn(II); (3) the intermediates encounter species 2, displacing the E89 side chain, and closing the interior channel, and the Mn(II) remains 6CN; (4) departure of all products from 6CN Mn(II) in species 4 re-establishes the 5CN starting material.

## Data Availability

Data from the HDX-MS experiments are provided as Appendix A.

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
