# Peer review of "Long-Range Allosteric Communication Modulated by Active Site Mn(II) Coordination Drives Catalysis in Xanthobacter autotrophicus Acetone Carboxylase"

_ijms, 2025, doi:10.3390/ijms26135945_

Round 1
Reviewer 1 Report
Comments and Suggestions for Authors
In this manuscript, the authors propose a significant conformational change, supported by crystal structures, that regulates the opening and closing of a tunnel to facilitate the passage of reaction intermediates between the nucleotide-binding and substrate phosphorylation sites, and the Mn(II)-dependent acetoacetate formation site. Through in-depth analyses using EPR and HDX-MS, they characterize equilibrium ligand-bound states and site-specific amino acid variants to explore the conformational dynamics associated with catalysis. Their results demonstrate that communication between the phosphorylation and acetoacetate formation sites is mediated by the exchangeable carboxylate ligand αE89 at the Mn(II) site. This study provides a novel paradigm for understanding carboxylation reactions in biological systems. I recommend publication after revision.
- The difference in denaturation temperature (Tm) shown in Figure 3 is relatively small. It is recommended to repeat the differential scanning fluorimetry (DSF) experiments at least three times to ensure the accuracy of the results.
- Please standardize the formatting of αE89, αE89D, and αE89A throughout the manuscript. Multiple inconsistencies have been noted—for example, line 138 on page 4 uses "aE89," the caption of Figure 4 uses "aE89A," and the last paragraph on page 9 refers to "αD89" and "αA89."
- As shown in Figure 6, intermediates such as carboxyphosphate and phosphoenolacetone are able to coordinate with the Mn(II) active site. In this context, could the phosphate groups of ATP itself also participate in coordination with the Mn(II) site?
- The reference list contains multiple inconsistencies in the formatting of author names and journal information. A thorough revision is strongly advised to ensure uniformity and adherence to journal style guidelines.
Reviewer 2 Report
Comments and Suggestions for Authors
In this manuscript the catalytic site of Mn(II) in acetone carboxylate from Xanthobacter autotrophocus was explored using EPR and HDX-MS spectroscopy, showing that E89 from the alpha domain play a role in Mn coordination during the catalytic procedure. The research seems to be well conducted and analyzed, and the conclusions are supported by the experiments. One minor point, it will be nice if DSF will also be conducted on the E89D mutant for comparison and additional control. Also, can the authors add some references for other studies that such spectral behavior of EPR was obtained for Mn upon AMP binding.
